# A New Variant of the *aadE-sat4-aphA-3* Gene Cluster Found in a Conjugative Plasmid from a MDR *Campylobacter jejuni* Isolate

**DOI:** 10.3390/antibiotics11040466

**Published:** 2022-03-30

**Authors:** Pedro Guirado, Elisenda Miró, Yaidelis Iglesias-Torrens, Ferran Navarro, Susana Campoy, Tyler Scott Alioto, Jessica Gómez-Garrido, Cristina Madrid, Carlos Balsalobre

**Affiliations:** 1Department of Genetics, Microbiology and Statistics, School of Biology, University of Barcelona, Av. Diagonal, 643, 08028 Barcelona, Spain; pedro.guirado.frias@gmail.com; 2Hospital de la Santa Creu i Sant Pau, Institut d’Investigació Biomèdica Sant Pau (IIB Sant Pau), Sant Quintí 89, 08041 Barcelona, Spain; emiro@santpau.cat (E.M.); yiglesiast@santpau.cat (Y.I.-T.); fnavarror@santpau.cat (F.N.); 3Departament de Genètica i Microbiologia, Universitat Autònoma de Barcelona, 08193 Cerdanyola del Vallés, Spain; susana.campoy@uab.cat; 4CNAG-CRG, Center for Genomic Regulation (CRG), Barcelona Institute of Science and Technology (BIST), Baldiri i Reixac 4, 08028 Barcelona, Spain; tyler.alioto@cnag.crg.eu (T.S.A.); jessica.gomez@cnag.crg.eu (J.G.-G.); 5Universitat Pompeu Fabra (UPF), 08002 Barcelona, Spain

**Keywords:** *Campylobacter*, conjugative plasmid, antibiotic resistance, aminoglycoside

## Abstract

*Campylobacter jejuni* is a foodborne pathogen causing bacterial gastroenteritis, with the highest incidence reported in Europe. The prevalence of antibiotic resistance in *C. jejuni*, as well as in many other bacterial pathogens, has increased over the last few years. In this report, we describe the presence of a plasmid in a multi-drug-resistant *C. jejuni* strain isolated from a gastroenteritis patient. Mating experiments demonstrated the transference of this genetic element (pCjH01) among *C. jejuni* by plasmid conjugation. The pCjH01 plasmid was sequenced and assembled, revealing high similarity (97% identity) with pTet, a described tetracycline resistance encoding plasmid. pCjH01 (47.7 kb) is a mosaic plasmid composed of a pTet backbone that has acquired two discrete DNA regions. Remarkably, one of the acquired sequences carried an undescribed variant of the *aadE-sat4-aphA-3* gene cluster, providing resistance to at least kanamycin and gentamycin. Aside from the antibiotic resistance genes, the cluster also carries genes coding for putative regulators, such as a sigma factor of the RNA polymerase and an antisigma factor. Homology searches suggest that *Campylobacter* exchanges genetic material with distant G-positive bacterial genera.

## 1. Introduction

*Campylobacter jejuni* is a worldwide foodborne pathogen. It is the zoonotic agent with the highest reported incidence in Europe, with 220,682 confirmed human cases and 47 reported deaths in 2019 [1]. *C. jejuni* causes campylobacteriosis, a self-limiting gastroenteritis that can sporadically cause systemic infection and trigger neurological disorders, such as Guillain–Barré syndrome [2]. *C. jejuni* is a motile, Gram-negative, microaerophilic epsilon proteobacteria that are characterized as being thermophilic with optimal growth in a temperature range between 30 and 45 °C, approximately. It is commonly found in the gastrointestinal tract of birds, including poultry species. In fact, the most common transmission pathway to humans is the consumption of undercooked chicken meat. Nevertheless, its presence has also been detected in the gastrointestinal tracts of mammals, insects, water, soil, etc. [3]. 

Over the last few years, an increase in the prevalence of antibiotic resistance has been noticed in many bacterial pathogens, including *Campylobacter*. A European Food Safety Authority report from 2018/2019 indicated that 61.5, 47.2, 1.5, 0.8, and 0.3% of the *C. jejuni* isolated from humans are resistant to ciprofloxacin, tetracycline, erythromycin, amoxicillin/clavulanic acid, and gentamicin, respectively [4]. The appearance of multidrug resistance (MDR) among *Campylobacter* species can jeopardize the treatment of prolonged and severe cases of campylobacteriosis. The spread of antibiotic resistance among bacteria is often mediated by plasmids that act as vehicles for antibiotic resistance genes and can be mobilized by conjugation [5]. It should be noted that tetracycline resistance in *Campylobacter*, which was detected in 47.2% of the human isolates from the European Union in 2018/2019, is commonly encoded in plasmids [6]. Several plasmids carrying tetracycline resistance in *C. jejuni* have been characterized and their transfer by plasmid conjugation has been experimentally demonstrated. In a recent report, we described that the conjugation of several tetracycline resistance carrying plasmids was promoted at temperatures characteristic of the avian intestinal tract [7]. The role of plasmid conjugation in the transfer of resistance to antibiotics other than tetracycline has been poorly characterized in *Campylobacter*. In this report, we describe a *C. jejuni* plasmid, pCjH01, isolated from a gastroenteritis patient in Spain that confers resistance to tetracycline, kanamycin, and gentamycin. The sequence of the pCjH01 plasmid reveals high similarity with the well-described tetracycline-resistance-encoding plasmid, pTet [8,9], but carries an undescribed variant of the *aadE-sat4-aphA-3* gene cluster providing resistance to kanamycin and gentamycin.

## 2. Results and Discussion

### 2.1. C. jejuni Strain H01 Carries a Conjugative Plasmid Providing Resistance to Tetracycline, Kanamycin, and Gentamicin

*C. jejuni* strain H01 belongs to a *C. jejuni* collection characterized in terms of the population structure, antimicrobial resistance, and presence of virulence-associated genes [10,11]. H01, isolated from a human patient suffering from gastroenteritis, belongs to the singleton ST-441 sequence type, which has been isolated from diverse niches, including humans, chickens, dogs, and farm environments [12]. The H01 isolate displays the most common profile of prevalence of virulence genes in the mentioned collection—being *cdtA*^+^, *cdtB*^+^, *cdtC*^+^, *cadF*^+^, *ciaB*^+^, *htrA*^+^, *hcp*^−^, *wlan*^−^, and *cgtB*^−^—whereas it has a distinct antibiotic resistance profile among the 150 isolates of the collection. The H01 isolate is a multi-drug-resistant (MDR) strain, being resistant to tetracycline (TET), nalidixic acid (NAL), ciprofloxacin (CIP), kanamycin (K), and gentamicin (G), and sensitive to ampicillin, amoxicillin/clavulanic acid, imipenem, streptomycin, chloramphenicol, and fosfomycin. 

In *Campylobacter*, resistances to TET, K, and G, all of which coincide in H01, are often encoded in plasmid-borne genes [6,13,14]. The *tet*(O) gene is the most common genetic determinant involved in resistance to TET, and the *aph(3′)-IIIa* gene is the one involved in K and G resistances [15,16]. The presence of these genes in the genome of the H01 strain, including both chromosomal and extrachromosomal elements, was confirmed by PCR and sequencing using specific primers, as described in Appendix A (data not shown). To determine whether these genes were located in a plasmid, S1 nuclease digestion of the genomic DNA was carried out, followed by pulsed field gel electrophoresis (PFGE) and the hybridization with specific probes for *tet*(O) and *aph(3′)-IIIa*. As a control, the genomic DNA of the strain 81-176 carrying the *tet*(O)-harboring plasmid pTet was also analyzed. Figure 1 shows that the S1 nuclease linearizes both plasmids, showing a band between 48.5 and 97 kb. The pCjH01 and pTet plasmids hybridize with the *tet*(O) probe, whereas only pCjH01 hybridizes with the *aph(3′)-IIIa* probe. These results clearly indicate that the resistance to TET, K, and G in H01 is conferred by the presence of a plasmid, from this point referred to as pCjH01. 

Conjugation is a widespread mechanism for the transfer of plasmids among bacteria, including *C. jejuni* [17]. To determine whether the putative plasmid present in strain H01 encodes a functional conjugative apparatus, conjugation experiments were performed. Mating assays were conducted at 42 °C for 4 h using A3S, a streptomycin (S)-resistant *C. jejuni* strain, as a recipient strain [7]. Transconjugants were selected in the presence of S and TET. Plasmid DNA from H01 and TET^R^ transconjugants in A3S was isolated, digested with *Bgl*II, and the restriction pattern characterized by electrophoresis. As it can be seen in Figure 2A, the digestion of the plasmid preparation from the H01 isolate displayed a specific restriction band profile, which was identical to the profile derived from the plasmid preparation of one TET^R^ transconjugant (Tet^R^ and S^R^ colonies). The selected TET^R^ and S^R^ colonies were confirmed as transconjugants by the PCR genotyping of the chromosomal *wlaN* gene, coding for a galactosyltransferase, since the A3S recipient strain, but not the H01 donor strain, carries the *wlaN* gene (Figure 2B). Hence, transconjugant clones were characterized by being S^R^, *wlaN*^+^, and carrying the pCjH01 plasmid (TET^R^). The selected transconjugants were also resistant to K and G, clearly indicating that the resistance to the three antibiotics was encoded in plasmid pCjH01. Conjugation rate for pCjH01 at 42 °C was estimated to 5.5 × 10^−7^ (Figure 2C). 

In a previous report, we showed that the conjugation of several *C. jejuni* plasmids was thermodependent, being higher at 42 than at 37 °C, the temperatures of the avian and human intestinal tracts, respectively [7]. Mating experiments with pCjH01 were also performed at 37 °C and similar conjugation rates to those at 42 °C were detected, indicating that pCjH01 conjugation is not temperature-dependent under the conditions tested. 

### 2.2. pCjH01 Is Derived from the pTet Plasmid and Carries a New Variant of the aadE-sat4-aphA-3 Gene Cluster

The plasmid pCjH01 was sequenced and assembled (Figure 3A) (GenBank accession number ON007234). It had a size of 47,758 bp and a GC content of 29.6 %. An in silico *Bgl*II restriction map was generated and, despite some deviation in the size estimation after electrophoretic analysis, the theoretical and experimental generated fragments coincided (Figure 2A), clearly confirming the presence of a single plasmid in strain H01. The annotation revealed 58 putative open reading frames (ORFs), covering approximately 94% of the plasmid (Table 1, Figure 3A). A BLAST alignment indicated that the pCjH01 plasmid had a high degree of identity with plasmids from *C. jejuni* and *C. coli*, including the pTet plasmid that has been previously characterized. The main known feature of the pTet plasmid is the presence of a canonical *tet*(O) gene that confers tetracycline resistance. The pCjH01 plasmid harbors a nearly identical *tet*(O) gene with only 11 nucleotide differences as compared with the *tet*(O) encoded in pTet, including a silent substitution in codon 21 and missense alterations in nine codons causing the following amino acid substitutions: P33L, K38E, Q142R, C295Y, L346I, E363G, K366T, G371C, and C595Y.

A comparison of the pCjH01 and pTet plasmid sequences (Figure 3A) revealed that both plasmids are related by shared homology over most of the sequence: 89% of pCjH01 aligns to pTet with 97% identity (with 5.8% of pTet unaligned), and includes most genes related to plasmid mobilization. Clear differences among the two plasmids were also detected, specifically within three distinct sequence regions (Figure 3B): region I (R1), where the pCjH01 plasmid carries a large sequence tract (4011 bp) that replaces a 1564 bp tract of the pTet plasmid; region II (R2), where the pCjH01 plasmid carries a sequence tract of 1163 bp that replaces a 749 bp sequence of pTet; and region III (R3), where, as compared to the pTet plasmid, it seems that a 288 bp deletion occurs. 

The large pCjH01 specific sequence (region I), from position 3292 to 7302, had a higher GC content (39.6%) than the plasmid on whole (29.6%), suggesting that this sequence was acquired in a later event. Homology searches revealed that part of region I (1–858 and 1793–4011) was found in several *Campylobacter* sequences located in both the chromosome and plasmids, such as the plasmid pGB19 (Figure 3A, Appendix A). pGB19 (CP071593.1) is a plasmid from *C. jejuni* strain GB19 associated with gastrointestinal infections linked to the onset of Guillain–Barré syndrome [18]. Interestingly, significant similarity was also found with sequences from distant microorganisms, such as *Mycoplasma bovirhinis* (AP018135.1) and *Streptococcus pyogenes* (CP010449.1) (Appendix A). 

The pCjH01-specific region I sequence carried six putative ORFs (Table 1, Figure 3B). Four of them (*g1776* to *g1779*), transcribed in the same orientation, encoded proteins putatively involved in antibiotic resistance, consistent with the antibiotic resistance phenotype associated with the acquisition of the pCjH01 plasmid. The other two ORFs (*g1780–g1781*), transcribed in the opposite direction with respect to the previous ORFs, encoded proteins presumably involved in gene regulation.

The sequence from *g1776* to *g1779* showed a certain degree of similarity with the well-described *aadE-sat4-aphA-3* cluster [19] (Appendix A). Several variants of these gene clusters have been described within the chromosome of *Campylobacter,* as well as transposons, plasmids, and phages of both G-positive and G-negative bacteria [20,21,22,23,24]. The canonical *aadE-sat4-aphA-3* gene cluster encodes a 6′-aminoglycoside-adenyltrasferase (also named AAD(6′)), a N-acetyltransferase, and a 3′-aminoglycoside-phosphotransferase which confer resistance to streptomycin, the atypical aminoglycoside streptothricin/nourseothricin, and to kanamycin/gentamicin, respectively. The *g1776-g1779* gene cluster in pCjH01 exhibits a high degree of similarity with the *aadE-sat4-aphA-3* cluster in plasmid pGB19 (Figure 4A), despite some important sequence rearrangements that will be further discussed below. The pGB19 *aadE-sat4-aphA-3* cluster is widespread among *Campylobacter* isolates and many other bacteria, including G-positive bacteria (Appendix A). The *g1776-g1779* cluster in pCjH01, though, seems to have been derived from an *aadE-sat4-aphA-3* that subsequently suffered drastic alterations in the DNA sequence: the pCjH01 cluster spanned 2630 bp in contrast to the 2047 bp of the pGB19 cluster. The homology among both clusters abruptly vanished within the *sat4* ORF, from codon 75 of 181. A sequence of 935 bp with no homology in the DNA databases replaced a 352 bp sequence of pGB19 carrying the 3′ of the *sat4* ORF and 34 bp of the intergenic region between *sat4* and *aphA-3* from pGB19 (Figure 4B). The 935 bp insertion generated a putative chimeric ORF (*g1777*) carrying the first 72 codons of *sat4* plus 39 extra codons. Moreover, the inserted sequence encoded an extra putative ORF (*g1778*) of 630 bp. The nucleotide sequence of *g1778* only had resemblance with sequences obtained from uncultured bacteria (KU544893). Homology searches indicated that the *g1778* predicted amino acid sequence had similarities with proteins from G-positive bacteria with phosphotransferase activity and involvement in aminoglycoside modification (Appendix A).

The two other ORFs present in pCjH01-specific region I encoded for a putative protein sharing high homology with an RNA polymerase sigma70 subunit (ORF *g1781*) and a protein carrying a zinc finger HC2 domain that was described in antisigma factors (ORF *g1780*). There was no intergenic region between *g1781* and *g1780,* suggesting that the two ORFs were cotranscribed. The 1363 bp sequence from the end of the *aphA-3* ORF to the end of region I, carrying both *g1781* and *g1780,* has been found in many *C. jejuni* and *C. coli* isolates in both plasmid and chromosome locations, as well as in several G-positive bacteria, such as *Streptococcus*, *Staphylococcus,* and *Enterococcus* (Figure 4A and Appendix A). 

Sigma subunits are responsible for promoter recognition and transcription initiation. Their expression is often regulated at the protein level by repressor proteins, known as antisigma subunits [25]. The presence of genes coding for a specific sigma/antisigma system in the plasmid pCjH01 suggests that some plasmid genes will require the specific sigma subunit for their correct expression. Considering the specific location of these sigma/antisigma systems in pCjH01 and that pTet does not carry such genes, we suggest that these proteins may control the expression of genes that are specifically present in the plasmid pCjH01, such as the genes involved in antibiotic resistance. Homology searches indicated that the genes present in the pCjH01 region I were found in phylogenetically distant organisms, including firmicutes (Appendix A). Genes acquired from distant organisms might not be properly expressed in the new host due to a lack of promoter recognition by the transcriptional machinery. The acquisition of a sigma subunit might allow the expression of the newly acquired genes. One can also expect that the sigma/antisigma system encoded in the pCjH01 is not restricted to controlling the expression of plasmid-encoded genes and could be involved in crosstalk between the plasmid and the chromosome if the sigma subunit from plasmid pCjH01 can recognize chromosomally encoded genes. Remarkably, genes highly similar to g1780 and g1781 have been found within a chromosomal multidrug resistance genomic island in the *C. coli* strain SH96 [26]. Sigma factors encoded in plasmids are rare. The plasmid pBS32 from *B. subtilis* encodes a sigma factor that is expressed under certain environmental stresses causing cellular death after DNA damage [27]. Interestingly, when using a longer sequence of pCjH01—spanning from the end of *aphA-3* to the start of *cpp7*—including the pTet sequences where the gene cluster was inserted, only pGB19 showed 100% homology in a correlative sequence (Appendix A). The identical relative location of the gene cluster in both plasmids suggests that they both derive from an ancestral pTet plasmid that acquired a gene cluster carrying the *aadE-sat4-aphA-3* cassette and the sigma and antisigma subunits, most probably from G-positive bacteria. This ancestral plasmid seems to have evolved in different ways: (i) it has acquired sequences coding for antibiotic resistances (pCjH01), and (ii) it has suffered severe rearrangements with other circulating plasmids generating mosaic plasmids, such as pGB19, that contain part of the ancestral pTet in a distinct plasmid backbone. A total of 96% of pCjH01 aligns to pGB19 with 96.24% identity (10.5% of pGB19 is unaligned); 83% of pGB19 aligns to pTet with 95.9% identity (10.6% of pTet is unaligned).

The pTet boundaries where the pCjH01-specific region I was inserted were the intergenic region between *cpp3* and *cpp4* and within the *repA* ORF, 146 bp upstream of the *repA* end. Therefore, the region I pCjH01-specific sequence replaced *cpp4* and part of *repA* from pTet. The *cpp4* ORF encodes a hypothetical protein that has domains that have been attributed to relaxase/mobilization nuclease domain-containing proteins [28]. RepA shares high homology with several proteins involved in replication. In previous reports, it has been shown that the *repA* gene can be a target for insertions [26].

A unique ORF, *g1790*, seems to be found in the pCjH01-specific region II (Figure 3A). Homology searches revealed that *g1790* encoded for a putative HTH transcriptional regulator from the XRE family. The presence of genes coding for XRE-type transcriptional regulators has been described in several conjugative plasmids. For instance, the pLS20 plasmid of *Bacillus subtilis* encodes an XRE regulator that represses plasmid conjugation [29,30], and the pRet42a plasmid from *Rhizobium etli* encodes an XRE-type protein that is required for conjugation from specific hosts [31]. These data suggest that the acquisition of this gene might provide differential regulation of plasmid conjugation affecting the optimal physiological and environmental conditions promoting pCjH01 conjugation and/or the range of specific hosts during conjugation. In a previous report, we showed that the conjugation of pTet and pTet-related plasmids are affected by temperature, being promoted at temperatures higher than 42 °C as compared to 37 °C [7]. Interestingly, pCjH01, despite sharing with pTet most of the sequence, including genes involved in plasmid transfer, did not seem to be temperature-dependent. Whether the regulator encoded in *g1790* is involved in the temperature regulation of conjugation remains to be elucidated. The acquisition of *g1790* causes the deletion of the *cpp16* ORF. Homology searches with the nucleotide sequence revealed that this gene was found in many *Campylobacter* isolates, located both in chromosomes and plasmids. Using the deduced amino acid sequence, it seemed to encode for a putative Gcn5-related N-acetyltransferase (GNAT). Acetyltransferases are enzymes involved in multiple bacterial processes, such as metabolism, cell wall modification, and antibiotic resistance. The role of this protein encoded in the pTet plasmid is uncharacterized. 

The last difference of note between the sequences of pCjH01 and pTet (region III) is the lack of a 288 bp sequence tract of pTet (from 5630 to 5917 bp, AY394561) in pCjH01 (Figure 3B). The deletion of this sequence causes alteration in two pTet ORFs: *cpp7* (5346–5726, 126 amino acids) and *cpp8* (5868–6266, 112 amino acids), with unknown functions. As a result, two new variants of the predicted *cpp7* and *cpp8* were generated (Figure 3B), which had no homology with any annotated protein. 

## 3. Conclusions

The spread of MDR among pathogens is a public health concern of capital relevance. *Campylobacter* is among the bacterial pathogens causing high incidence of infections in humans, particularly in certain geographical areas such as the EU, North America, and Australia [32]. Understanding the mechanisms of antibiotic resistance and its mobilization among bacteria is of crucial importance to design strategies aiming to avoid the spreading of antibiotic resistances [33]. In this report, we describe pCjH01, a plasmid from a *C. jejuni* isolated from a patient with gastroenteritis that confers resistance to tetracycline, kanamycin, and gentamycin. pCjH01 has particular and distinctive characteristics. We demonstrate that pCjH01 is self-transmissible by conjugation and it seems to evolve from the very well-known pTet plasmid, characterized to transfer resistance to tetracycline, since it shares a high percentage of identity. The main distinctive features of pCjH01 are the gain of a gene cluster encoding for antibiotic resistance and a sigma/antisigma system, the gain of a gene encoding for a transcriptional regulator, and the loss of some ORFs. Homology searches using pCjH01-specific sequences revealed high similarity with sequences from G-positive bacteria, indicating that *Campylobacter* exchanges genetic material with distant bacterial species, such as *Staphylococcus* and *Streptococcus*. Evidence of gene exchange between *Campylobacter* and G-positive bacteria has been proposed earlier [14,24]. Whether the exchange occurs via natural transformation or conjugation remains unclear. 

The pCjH01 plasmid carries a new variant of the *aadE-sat4-aphA-3* gene cluster. In pCjH01, the *aadE* and *aphA*-3 genes were intact, as compared to the *aadE-sat4-aphA-3* gene cluster present in pGB19 and many other *C. jejuni* and *C. coli* isolates. In contrast, the downstream sequence of *sat4* was substituted, generating a chimeric protein of smaller size (13.4 kDa as compared to the 21.3 kDa of *sat4*) with an unknown function. Moreover, the pCjH01 variant of the *aadE-sat4-aphA-3* gene cluster carried an extra ORF encoding a putative phosphotransferase that may have been involved in antibiotic resistance. In many cases, the *aadE-sat4-aphA-3* gene cluster was found together with two genes encoding for putative sigma and antisigma subunits, as exemplified by the plasmid pGB19 (Figure 4A). Homology searches using the genetic element including both the *aadE-sat4-aphA-3* gene cluster and the sigma and antisigma genes indicated that several variants of the genetic environment existed when compared to pGB19, for instance, in the pN29710 plasmid and related (Figure 4B and Appendix A). To our knowledge, the previously detected variations do not affect directly the *aadE-sat4-aphA-3* gene cluster, but the genomic environment, whereas, in the pCjH01 plasmid, the *aadE-sat4-aphA-3* gene cluster was altered by the insertion of a new gene. Further studies will be required to functionally characterize the newly described variant of the *aadE-sat4-aphA-3* gene cluster and to determine its spread among bacterial pathogens. 

## 4. Materials and Methods

### 4.1. Bacterial Strains, Plasmids, and Growth Conditions

The *C. jejuni* strains H01 and 81-176 were isolated from human patients suffering from campylobacteriosis. The strain H01 was previously isolated by our research group [10]. The strain 81-176 carries the plasmids pVir and pTet, conferring the pTet plasmid resistance to tetracycline [9]. The *C. jejuni* A3S strain is nalidixic acid-resistant, streptomycin-resistant, and tetracycline-susceptible [7], and it was used as the recipient strain during the mating experiments. All *C. jejuni* strains were grown on Columbia blood agar base (CBA, Oxoid) supplemented with 5% of defibrinated sheep blood (Oxoid, Hampshire, UK). When required, culture media was supplemented with tetracycline (TET), kanamycin (K), gentamicin (G), nalidixic acid (Nal), and streptomycin (S) at 20, 25, 10, 50, and 15 µg/mL, respectively. CBA plates were incubated for 48 h at either 37 or 42 °C under microaerophilic conditions using a CampyGen^TM^ atmosphere generation system (Oxoid, Hampshire, UK).

### 4.2. Mating Experiments

Mating experiments were performed essentially as described earlier [7]. Donor and recipient strains were grown for 48 h at either 37 or 42 °C on CBA plates supplemented with tetracycline and streptomycin, respectively. Bacterial cells were collected in phosphate-buffered saline (PBS) supplemented with MgCl_2_ (100 µM), and the OD_550_ of the cell suspension was normalized to 1.5. Equivalent volumes (50 µL) of recipient and donor cell suspensions were mixed in the presence of DNAse I (100 U/mL) (Roche, Basel, Switzerland) to avoid natural transformation events during the assay. Aliquots of 15 µL were spotted on CBA plates supplemented with DNAse I (100 U/mL) and incubated at either 37 or 42 °C for four hours. Cells were recovered in PBS, serially diluted, and spread on CBA plates supplemented with the required antibiotics for the selection of both donor (tetracycline) and transconjugant (tetracycline plus streptomycin) cells. Control mating experiments with only donor or recipient cells were included in all experiments. Plates were incubated for 48 h at 42 °C and the frequency of conjugation was calculated as the number of transconjugants per donor cell. We show the results of at least three independent experiments in a scatter dot plot graphic with the average. Plasmid isolation, restriction profile characterization, and strain genotyping by the detection of the *wlaN* gene were used to confirm transconjugant selection. 

### 4.3. DNA Techniques

Genomic DNA (total DNA content in the cells) and plasmid DNA were extracted using InstaGene^TM^ Matrix (Bio-Rad, Hercules, CA, USA) and E.Z.N.A.^®^ Plasmid DNA kit (Omega Bio-tek, Norcross, GA, USA), respectively, following supplier indications. DNA was isolated from bacterial cultures grown at 42 °C for 48 h on CBA plates. The *Bgl*II restriction pattern was used to characterize the isolated plasmids. PCR amplification was performed with PCR MasterMix (2×) (Thermo Scientific, Waltham, MA, USA). All the primers used are described in Appendix A.

### 4.4. S1-PFGE and DNA Hybridization

S1 digestion of genomic DNA was carried out as described previously in order to convert supercoiled plasmids into full-length linear molecules [34]. DNA samples were subjected to separation by PFGE using the CHEF-DR III Pulsed Field Electrophoresis Systems (Bio-Rad, Hercules, CA, USA) and following the standard operating protocol from PulseNet (https://pulsenetinternational.org/assets/PulseNet/uploads/pfge/PNL03_CampyPFGEprotocol.pdf (accessed on 27 February 2022)). The DNA within the PFGE gel was transferred onto a positively charged nylon membrane and UV light cross-linked by standard methods [35]. Specific digoxigenin-labeled probes for *tet*(O) and *aph(3′)-IIIa* genes were obtained using PCR (primer pairs indicated in Appendix A) and the digoxigenin PCR DIG Probe Synthesis kit (Roche, Basel, Switzerland). Southern blot hybridization was carried out under high-stringency conditions according to the manufacturer’s instructions.

### 4.5. Genome Sequencing, Plasmid Assembly, and Alignment

For the whole-genome sequencing of the *C. jejuni* isolates H01, the DNA was extracted using the Wizard DNA-Purification kit (Promega, Madison, WI, USA). Genomic libraries and sequencing were performed in ADM Life Sequencing (https://www.biopolis-microbiome.com/ (accessed on 27 February 2022)) using the Illumina NextSeq 500 platform and Nextera XT 150PE/TrueSeq DNA kit for library preparation. The pCjH01 plasmid was assembled from the trimmed reads with SPAdes v3.13.0 [35], with the plasmid option in addition to default parameters. 

The pCjH01 plasmid was assembled from the trimmed 150 bp paired-end reads (cutadapt with parameters: -f fastq -e 0.1 -q 10 -O 1 -a AGATCGGAAGAGC) with Unicycler v0.4.6 [36], which in turn ran SPAdes v3.13.0 [37], with default parameters. An assembly with two connected components was obtained. The smallest connected component consisted of two contigs with lengths of 47,651 and 107 bp, which, after inspection of the assembly graph, were further joined into a single contig of 47,758 bp. The origin was then rotated to begin with the *tet*(O) gene.

We annotated the assembled plasmid with Prokka v1.12 [38]. Resistance Gene Identifier (RGI), with CARD database version 3.0.2 [39] was used to predict resistomes from the genome assembly. The Prokka and RGI annotations were combined to produce a final protein-coding gene set. To compare the assembled plasmids with the reference plasmid pTet (accession: AY394561.1) as well as pGB19 (accession: CP071593), pairwise BLAST [40] alignments were performed (blastn e-value cutoff of 10^−5^) and additional pairwise comparisons were made with Mummer v3.22 [41] to determine percent identity. To assign gene names to their corresponding orthologue in pTet, we performed an all versus all Blastp with all of the protein sequences. Additional homologous plasmids were identified using the NCBI BLAST service against nt with default blastn parameters. To visualize the homology of pCjH01, pTet, and pGB19, we aligned the plasmids with Minimap2 [42] and parsed the alignments in order to construct a pangenome sequence of pTet and pCjH01 for use as reference with the CGView Comparison Tool [43]. The protein-level homology of the antibiotic resistance cassette of pCjH01 (region: 1–7156) to pGB19 and an additional plasmid, pN29710-1 (accession: CP004067), were compared with Clinker v0.0.23 [44].

## Figures and Tables

**Figure 1 antibiotics-11-00466-f001:**
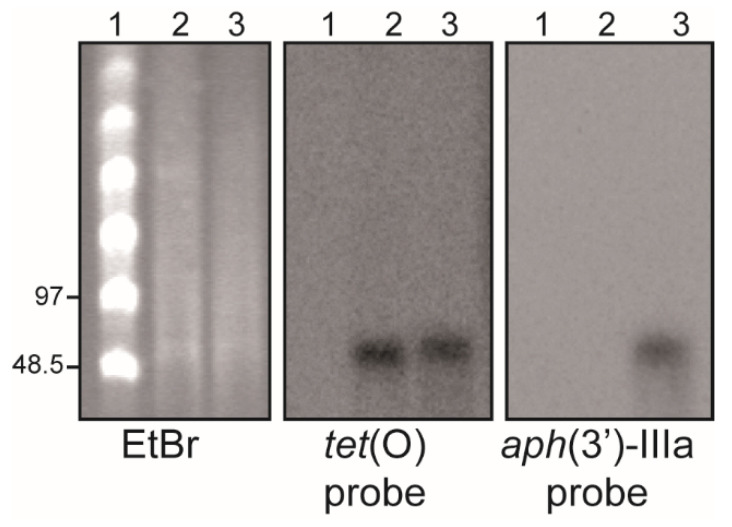
The HS01 isolate carries a MDR plasmid, pCjH01. PFGE-S1 hybridization with specific probes for *tet*(O) and *aphA-3* (*aph(3′)-IIIa*). Genomic DNA from the *C. jejuni* 81-176 strain carrying the pTet plasmid and the MDR H01 isolate was S1 digested, and linearized plasmids were separated by PFGE. The DNA was visualized by ethidium bromide (EtBr) staining and the presence of the *tet*(O) and *aphA-3* genes was detected by hybridization using specific probes, as indicated. Lane 1, molecular mass marker (CHEF DNA Size Standard, 48.5–1000 kb, Lambda Ladder, Bio-Rad). The size of some of the bands in kilobases is indicated on the left side of the EtBr panel. Lanes 2 and 3, S1 digested genomic DNA from 81-176 and H01, respectively.

**Figure 2 antibiotics-11-00466-f002:**
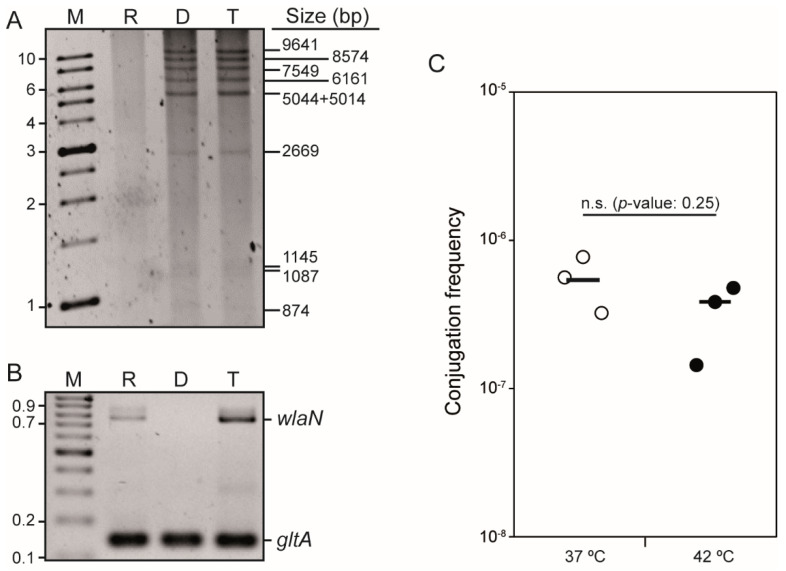
The pCjH01 plasmid from the H01 isolate is a conjugative plasmid that carries resistance to tetracycline, kanamycin, and gentamycin. (**A**) *Bgl*II restriction profile of DNA plasmid from recipient (R), donor (D), and transconjugant (T) strains. The R strain was A3S, and the D strain was H01. Lane M, molecular mass marker (1 Kb DNA Ladder, Nippon Genetics, Düren, Germany)—the size of some of the bands in kilobases is indicated on the left side of the panel. On the right side of the panel, the theoretical size in base pairs of the bands derived from the deduced *Bgl*II restriction map from the pCjH01 sequence obtained is indicated. The three lowest mass bands were better detected when the gel was overexposed (Appendix A). (**B**) Genotyping of the R, D, and T strains, as in A, by the PCR amplification of the *wlaN* gene. Lane M, molecular mass marker (GeneRuler 100 bp DNA Ladder, ThermoScientific, Waltham, MA, USA)—the size of some of the bands in kilobases is indicated on the left side of the panel. PCR detection of the housekeeping gene *gltA* was used as a control. (**C**) Conjugation frequency of pCjH01 at 37 and 42 °C using donor and recipient cells as in A. Both cultures and mating mixtures were incubated at the indicated temperatures. Significance was tested by an impaired two-tailed *t*-test. Statistical significance (*p*-value) is indicated, n.s. indicates no significance.

**Figure 3 antibiotics-11-00466-f003:**
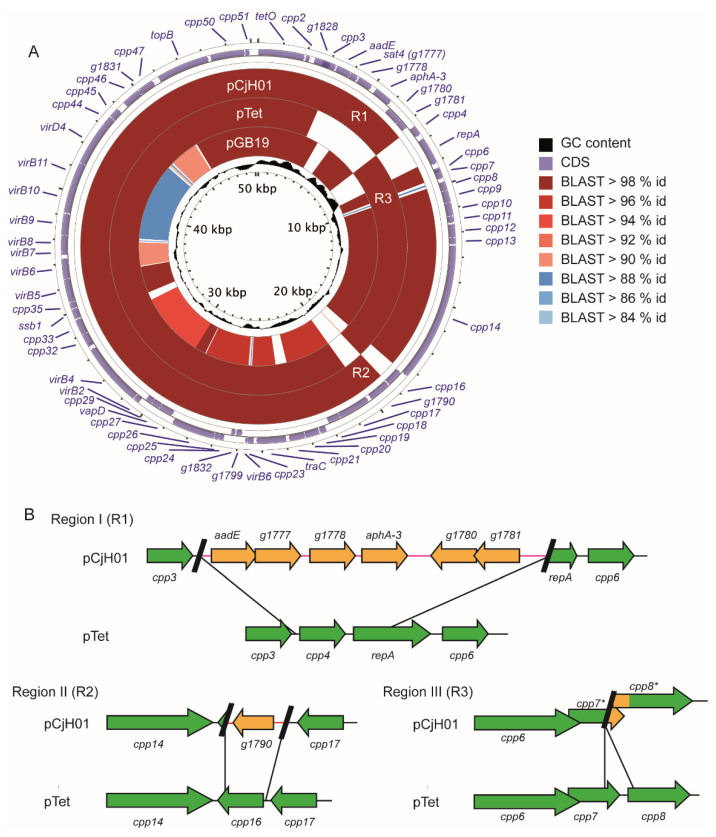
The pCjH01 plasmid has high similarity with the pTet plasmid. (**A**) Comparison between the pCjH01, pTet, and pGB19 plasmids. The regions I (RI), II (RII), and III (RIII) are indicated. The three plasmids were aligned by blastn to an inclusive artificial “pangenome” sequence constructed from both pCjH01 and pTet to show regions that are present or absent from each plasmid with respect to the others. The level of BLAST percent identity is indicated by color according to the legend. The annotated ORFs correspond to the combined annotation of all three plasmids. (**B**) Representation of regions I, II, and III in both pCjH01 and pTet. Green arrows and black lines indicate the sequences present in both plasmids, whereas orange arrows and red lines indicate the sequences only present in pCjH01.

**Figure 4 antibiotics-11-00466-f004:**
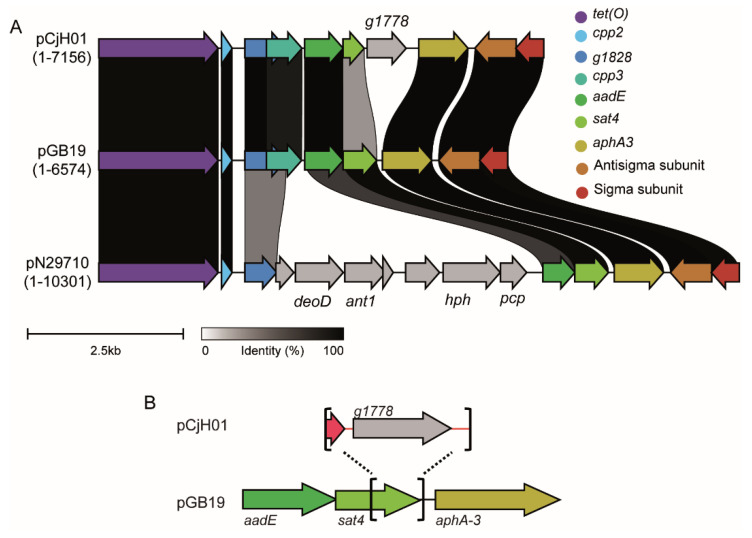
The new variant of the *aadE-sat4-aphA**-3* gene cluster of pCjH01. (**A**). Comparison between *aadE-sat4-aphA-3* gene clusters among the pCjH01, pGB19 (CP071593.1), and pN29710 (CP004067) plasmids. Homology at the protein level is indicated both with a color code according to the legend and ribbons indicate the pairwise percent identity with respect to pGB19. (**B**) Brackets and dotted lines demarcate the substitution of the 3′ *sat4* sequence and insertion of a new gene, g1778, in pCjH01 with respect to pGB19.

**Table 1 antibiotics-11-00466-t001:** ORFs annotated in the plasmid pCjH01. The pCjH01-specific ORFs, as compared to pTet, are highlighted (grey background).

Gene (ORF)	AA(MW)	Putative Function *
*tetO*	639 (72.5)	Tetracycline resistance ribosomal protection protein
*cpp2*	57 (6.7)	Conjugal transfer protein
*g1828*	221 (25.5)	Relaxase/mobilization nuclease-domain-containing protein
*cpp3*	190 (22.1)	Hypothetical protein
*aadE*	206 (24.3)	Aminoglycoside 6 adenyltransferase.
*sat4*	113 (13.4)	Streptothricin acetyltransferase
*g1778*	209 (23.9)	Aminoglycoside phosphotransferase
*aphA-3*	264 (31)	Aminoglycoside O phosphotransferase APH(3′)IIIa
*g1780*	221 (25.1)	Anti sigma factor (Zinc-finger domain containing protein)
*g1781*	150 (17.1)	Sigma 70 family RNA polymerase sigma factor
*cpp6*	58 (7.0)	Hypothetical protein
*cpp7*	96 (11.9)	Hypothetical protein
*cpp8*	117 (13.4)	Hypothetical protein
*cpp9*	170 (19.6)	Hypothetical protein
*cpp10*	185 (22.1)	Hypothetical protein
*cpp11*	88 (10.6)	Hypothetical protein
*cpp12*	186 (21.5)	ParA hypothetical protein
*cpp13*	88 (10.2)	Hypothetical protein
*cpp14*	1932 (224.3)	DEAD/DEAH box like helicase
*g1790*	314 (37.3)	HTH transcriptional regulator, XRE family
*cpp17*	462 (54.0)	Relaxase/mobilization domain containing protein
*cpp18*	183 (21.1)	Hypothetical protein
*cpp19*	93 (11.5)	Hypothetical protein
*cpp20*	203 (23.9)	NTPase. Hypothetical protein
*cpp21*	217 (26.0)	Hypothetical protein
*traC*	408 (47.1)	DNA primase, TraC family
*cpp23*	87 (9.7)	EexN family lipoprotein. IncN-type entry exclusion
*virB6*	32 (3.6)	Truncated VirB6 protein
*g1799*	85 (10.2)	Hypothetical protein
*g1832*	61 (7.4)	Hypothetical protein
*cpp24*	72 (8.1)	Type-II toxin-antitoxin system HicB family antitoxin
*cpp25*	67 (7.9)	Type-II toxin-antitoxin system HicA family toxin
*cpp26*	597 (69.0)	AAA family ATPase
*cpp27*	204 (26.6)	Recombinase family protein
*vapD*	125 (15.0)	Virulence associated protein
*cpp29*	107 (12.6)	Hypothetical protein
*virB2*	87 (9.2)	TrbC/VirB2 family protein. Conjugal transfer protein TraC
*virB4*	992 (106.3)	VirB4 family type IV secretion/conjugal transfer ATPase
*cpp32*	188 (21.8)	Rha family transcriptional regulator/phage regulatory protein
*cpp33*	221 (25.6)	Hypothetical protein
*ssb1*	141 (15.8)	single-stranded DNA-binding protein
*cpp35*	91 (10.7)	Hypothetical protein
*virB5*	323 (37.2)	Type IV secretion system protein
*virB6*	332 (35.5)	Type IV secretion system protein. TrbL/VirB6 family protein
*virB7*	56 (6.3)	Hypothetical protein
*virB8*	220 (25)	Type IV secretion system protein
*virB9*	295 (34.1)	P-type conjugative transfer protein VirB9
*virB10*	391 (43.1)	Type IV secretion system protein VirB10. TrbI/VirB10 family protein
*virB11*	330 (37.6)	P-type DNA transfer ATPase VirB11
*virD4*	603 (63.7)	Type IV secretory system conjugative DNA transfer family protein
*cpp44*	145 (16.8)	cag pathogenicity island protein
*cpp45*	254 (29.4)	Hypothetical protein
*cpp46*	265 (30.6)	Hypothetical protein
*g1831*	30 (3.9)	Hypothetical protein
*cpp47*	206 (23.8)	Hypothetical protein
*topB*	730 (84)	DNA topoisomerase III
*cpp50*	473 (57)	Hypothetical protein
*cpp51*	59 (7.1)	Hypothetical protein

* The predicted amino acids were used to search for putative function using the NCBI BLAST service with default blastp parameters.

## Data Availability

The data presented in this study are openly available in GenBanK data base (https://www.ncbi.nlm.nih.gov/genbank/, accessed on 1 March 2022) accession number ON007234.

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
