# Peer review of "A New Variant of the aadE-sat4-aphA-3 Gene Cluster Found in a Conjugative Plasmid from a MDR Campylobacter jejuni Isolate"

_antibiotics, 2022, doi:10.3390/antibiotics11040466_

Round 1

Reviewer 1 Report

The manuscript describes the presence of a plasmid in a multi drug resistance strain of C. jejuni.  Mating experiments were formed to verify that the plasmid harbored genes for antibiotic resistance. The pCjH01 plasmid was sequenced and assembled to show a high level of similarity with pTet.  A comparison of pTet and pCjH01 was carried out. 

Overall the manuscript is very well written and straight forward.  Experimental design and results are well described and thorough.  Referencing of literature in the manuscript is also thorough.

As a result, only minor revisions are requested. Specific comments are below

Specific comments:

line 44: define EFSA

line 52: define EU

Line 61: in this sentence, just explain the significance of the two temperatures (37 and 42 ºC) to the average reader.

Line 69: expand out, use "Gram-positive" instead of G+ 

Figure 1: what ladder was used?

Line 111: abbreviate Campylobacter jejuni to C. jejuni

Table 1: why are some of the rows shaded grey?

Line 149: define ORFs

Line 203: expand out G+ and G- to Gram-positive to Gram-negative

Line 28,  211, 240, 325: expand G+ to Gram-positive

Reviewer 2 Report

The manuscript of Pedro Guirado and coauthors titled "A new variant of the aadE-sat4-aphA-3 gene cluster found in a conjugative plasmid from an MDR Campylobacter jejuni isolate" is devoted to the study of new conjugative plasmid encoding antibiotic resistance in C. jejuni clinical isolate. The manuscript is well written and I only found one issue regarding the unavailability of depositing sequence data of pCjH01 plasmid in the public domain (GenBank/ENA).

The minor issues are the next one:

Line 3: Omit dot at the end of the title.

Line 16: Omit dash in foodborne

Line 22: I suggest adding the molecular weight of the plasmid in kb

Line 28 and so forth: I suggest replacing G+ and G- with gram-positive and gram-negative throughout the text of the manuscript.

Line 33: Omit dot in number

Lines 60-71: I suggest putting it in Conclusion, not the Introduction

Line 88: Replace with "TET, K and G"

Line 93: Omit "(see Materials and Methods)"

Lines 107 and 134: Which molecular mass marker was used?

Figure 2A: I can't see anything with sizes 1,145, 1,084, and 874 in R, D, or T?

Table 1: What means colored lines?

Line 308: I suggest using Conclusion

Line 339: Where is Table S9?

Line 380: Omit TM

Line 384: What PFGE apparatus was used?

Line 395: What model of NextSeq was used?

Line 395: Replace Nextyera with Nextera

Line 410: Replace 10-5 to 10^-5 or 10e-5

Line 419: Version of clinker?

Author Response

Please see he attachment 

Reviewer 3 Report

In this manuscript, the authors describe a new antibiotic-resistance gene cluster aadE-sat4-aphA-3 existed in a conjugative plasmid from a MDR C. jejuni isolate. Gene comparison was conducted among many relevant species’ genomic and plasmids genes. Please double check writing throughout the manuscript. The following are the specific concerns.

  1. At line 44: Write the full name instead EFSA.
  2. At line 88: Instead to write TET and K, G resistance, write TET, K and G resistance.
  3. At line 89 “The presence of these genes in the genome of the H01 strain was confirmed by PCR”, I want to know how you separate genomic and plasmid DNA? And if you statement here is correct, which means genomic DNA also has antibiotic-resistant gene to TET, K, and G, then why at line 97-99, you got this conclusion? “These results clearly indicate that the resistance to TET, K and G in H01 is conferred by the presence of a plasmid, from this point referred to as pCjH01.”.
  4. At line 91: The S1 is indicated for what?
  5. At line 96: What is the size for tet(O) and aph(3’)-IIIa genes?
  6. At line 102: What is HS01?
  7. At line 111: Instead to write Campylobacter jejuni write jejuni.
  8. At line 113-114, please rephrase this sentence “using as a recipient strain the streptomycin (S) resistant strain A3S”.
  9. In line 121: You should mention wlaN gene is carry which characteristic.
  10. Figure2 A and B, which marker you used? Please write the marker name and its company name. In figure 2B, what does gltA mean?
  11. At line 132, the A shouldn’t be bolded.
  12. Figure 3A, you should mark the borders of R1,R2,R3.
  13. At line 196, it’s not necessary to put Table 1 in the main manuscript.
  14. At line 200 and 208, “shows certain degree of similarity”, you’d better change certain to an exact number. And Table S3 shows sequences from different source with similarity of pCjH01-sequence carrying from g1776 to g1779. In my opinion, if you want to prove “The sequence from g1776 to g1779 shows certain degree of similarity with the well described aadE-sat4-aphA-3 cluster”, you should compare the sequence between g1776 to g1779 and aadE-sat4-aphA-3 cluster.
  15. In figure 4A, the information of pN29710 should be provided. Is that from uncultured bacteria (KU544893)?
  16. The sentence at line 311 needs a citation.
  17. At line 350, A3S strain is nalidixic acid and streptomycin resistant, and tetracycline susceptible, you should mention it’s susceptible to tetracycline, because that’s the properties of this strain that can be used as a recipient.
  18. From the paper you cited, I know A3S is also C. jejuni, but you’d better mention it here?
  19. And at line 347, you’d better say “The two jejuni strains”. Because we don’t know A3S isolated from patient or not.
  20. At line 358, the matting assay is not clear stated, you should give the information of what plasmid (with antibiotic-resistant genes) the donor has, what antibiotic-resistant genes the receptor has on its chromosome, and what characteristics will the transconjugants Usually, people perform matting assay will use three plates, one with the antibiotic that donor can grow, one with antibiotic that receptor can grow, another with both of the antibiotics that transconjugants can grow.
  21. At line 359-360, for donor and recipient strains, they use different antibiotics as selective marker, it should be clarified which antibiotic plate you used for each strain. Same for the culture time, 42 for donor ( jejuni), 37 for recipient(A3S), the order is wrong in the paper.
  22. At line 362, the most common way to assess microbial growth in solution is the measurement of the optical density at 600 nm, or OD600.
  23. At line 367, what’s the “required antibiotics”, please be clarified.
  24. At line 375, you’d better briefly describe the method instead of just put their names there.
  25. At line 383: At least write a summary about it.
  26. At line 382, the name of this method is usually called “S1-PFGE and DNA hybridization”.
  27. At line 385, please change to “the DNA on the PFGE gel transfer onto”. The link is not working (Website unavailable).
  28. Line 396-397 and line 398-400 have some repeat contents.
  29. In line 415: The plasmid pGB19 from where?

Round 2

Reviewer 2 Report

The authors have eliminated all the flaws I mentioned. I hope that plasmid sequencing data will available soon in NCBI GenBank. The manuscript may be accepted for publication.

Reviewer 3 Report

None